# Association between Urban Greenness and Depressive Symptoms: Evaluation of Greenness Using Various Indicators

**DOI:** 10.3390/ijerph16020173

**Published:** 2019-01-09

**Authors:** Hyeonjin Song, Kevin James Lane, Honghyok Kim, Hyomi Kim, Garam Byun, Minh Le, Yongsoo Choi, Chan Ryul Park, Jong-Tae Lee

**Affiliations:** 1Urban Forests Research Center, National Institute of Forest Science, Seoul 02455, Korea; etaoin0810@naver.com (H.S.); maeulsoop@korea.kr (C.R.P.); 2School of Public Health, Boston University, Boston, MA 02118, USA; klane@bu.edu; 3BK21PLUS Program in ‘Embodiment: Health-Society Interaction’, Department of Public Health Science, Graduate School, Korea University, Seoul 02841, Korea; honghyok@korea.ac.kr (Ho.K.); ippiy@hanmail.net (Hy.K.); garam0110@gmail.com (G.B.); 91.minhle@gmail.com (M.L.); yschoi860@gmail.com (Y.C.); 4School of Health Policy and Management, College of Health Science, Korea University, Seoul 02841, Korea

**Keywords:** urban greenness, depressive symptoms, logistic regression, Normalized Difference Vegetation Index

## Abstract

An increasing number of studies have suggested benefits of greenness exposure on mental health. We examined the association between urban greenness and depressive symptoms in adults in the general population living in the seven major cities in Korea (*N* = 65,128). Using data from the Korean Community Health Survey 2009, depressive symptoms were measured on the Center for Epidemiological Studies Depression Scale (CES-D). Greenness was assessed using Normalized Difference Vegetation Index (NDVI) and land-use data (forest area and forest volume). Logistic regression models were fitted to adjust for potential confounders. Individuals in regions with the highest NDVI (quartile 4) had the lowest odds for depressive symptoms compared to quartile 1, after adjusting for potential confounders (OR = 0.813; 95% CI: 0.747, 0.884). For all greenness indicators except for forest area per district area (%), the highest rate of depressive symptoms was found for the individuals in the lowest quartile of greenness (quartile 1) and the lowest rate of depressive symptoms for those in the highest quartile of greenness (quartile 4). We found an inverse association between urban greenness and depressive symptoms, which was consistent across a variety of greenness indicators. Our study suggests health benefits of greenness and could provide a scientific basis for policy making and urban planning.

## 1. Introduction

Depressive disorders are known to impose burdens on the population such as functional burdens due to cognitive impairment, social burdens due to decreased productivity, and higher mortality due to suicide [1]. According to the Global Burden of Disease Study 2015, depression is the 2nd leading causes of global age-specific disability-adjusted life years (DALYs) in 2015 for aged 20–24 years, the 3rd for aged 15–19, and between 5th and 10th for aged 40–64 years [2].

Urban green space is one of the most important environmental components of a city in that it provides health benefits to the city dwellers. Exposure to greenness is known to have a positive impact on health through various pathways including the promotion of physical activity, enhancement of social interaction, and mitigation of noise and air pollution [3,4,5]. Beyond these pathways, people can also benefit from green space through its inherent protective effect on health [6,7]. Based on the biophilia hypothesis and psycho-evolutionary theory claiming that humans have an inherent need of affiliation with nature and that natural environments bring emotional stability and help with faster recovery, it has been suggested that exposure to greenness could directly grant people restorative health effects [8,9]. Based on this explanation, a number of epidemiological studies have explored the relationship between greenness exposure and mental disorders such as depression, anxiety, and psychological distress. Depressive disorders were assessed by various methods including psychological measurement tools, hospital visits and medication. In the cross-sectional study in England, depressive symptoms were assessed through General Health Questionnaire and found that pregnant women in the greener area were 18–23% less likely to have depressive symptoms than those in the least green quintile [10]. Maas et al. used medical records on diagnosis of depression to estimate the prevalence of depression and found that the prevalence of depression was lower in greener areas [11]. Helbich et al. found an inverse association between green space and antidepressant prescription rates for the individuals in the Netherlands [12]. For assessing greenness exposure, land-use data and satellite imagery have been used for measuring the residential distance to green space, amount of green space, and level of greenness. Beyer et al. assessed greenness exposure through Normalized Difference Vegetation Index (NDVI) which is a satellite-based data and tree canopy coverage derived from land-use data and found lower levels of depressive symptoms in the greener area [13]. A protective effect of greenness on mental health has generally been found, but the study areas were limited to Europe, North America, and Australia [4,5,14].

In South Korea, mental health problems have been some of the most serious issues. According to the Organization for Economic Co-operation and Development (OECD), the suicide rate of South Korea has continuously ranked within the top two among the 35 OECD countries since 2003 [15]. With the growing concerns for the environment and the burden of depressive disorders in Korea, understanding the association between environmental factors and mental health is required to reduce such burden. However, research on the association between greenness exposure and mental health in Korea is very limited. The objective of this study was to examine the association between greenness exposure and depressive symptoms in adults living in the seven major cities in Korea, using various greenness indicators. We hypothesized that higher level of greenness level would be associated with lower rate of depressive symptoms.

## 2. Materials and Methods

### 2.1. Study Area and Study Population

This study was based on the Korean Community Health Survey (CHS) 2009 conducted from September to November of 2009. The CHS is a cross-sectional survey of Korean adults over 19-year-old that has been conducted annually since 2008 by the Centers for Disease Control & Prevention in Korea [16]. The main purpose of the CHS is to fulfill the need for standardized health statistics across Korea and to provide a basis for public health policymaking. The CHS was administered via Paper Assisted Personal Interviewing (PAPI) through home visits by trained staff and collected data on socio-economic status (SES), lifestyle, and health status.

The CHS 2009 surveyed 227,700 individuals throughout Korea, but our study restricted the study population to the residents of the seven major cities of Korea (Seoul, Busan, Daegu, Incheon, Gwangju, Daejeon, and Ulsan) to measure the effect of urban greenness. Among the 68,647 residents of the seven major cities that completed the CHS, 251 individuals who did not finish the Center for Epidemiological Studies Depression Scale (CES-D) section which measures depressive symptoms were excluded. An additional 3268 individuals lacking information on covariates including SES and health status were excluded. 85% of these individuals did not have household income data. There was no difference in CES-D score between the individuals who lacked household income data and the individuals who did not.

The total study population was 65,128 adults living in the seven major cities in Korea. The residential geographical codes of the study population were district-level and used for matching exposure data. A district is an administrative unit that sub-divides a city in Korea with a total 74 districts across the seven major cities.

### 2.2. Assessment of Depressive Symptoms

The CHS 2009 measured depressive symptoms using the Korean version of the CES-D. The CES-D was developed by Radloff for the purpose of measuring current levels of depressive symptoms in the general population and consists of 20 items asking one to mark the frequency of each symptom over the past seven days [17]. Each item is rated on a 4-point Likert scale, and the total score ranges from 0 to 60. A higher score indicates an elevated risk for depression and the cut-off score of ≥16 is often used for identifying individuals at high risk of depression [18]. In this study, individuals who had a CES-D score greater than or equal to 16 were defined as having depressive symptoms.

### 2.3. Exposure Assessment

Normalized Difference Vegetation Index (NDVI) is a measure of photosynthetic activity and has been used in many studies as a measure of greenness. NDVI was obtained from the Moderate Resolution Imaging Spectroradiometer (MODIS) sensor on the National Aeronautics and Space Administration (NASA)’s Terra satellite as 16-day composites at 250 m × 250 m resolution. The MODIS NDVI product used is the MOD13Q1 (National Aeronautics and Space Administration (NASA)/U.S. Geological Survey (USGS) 2015) Version 6 data, which have been corrected for atmospheric contamination from water, clouds, and aerosols, and will hereafter be referred to as greenness. A total of 23 satellite images of the Korean peninsula from 31 October 2008 to 16 October 2009 were acquired and used to calculate the annual average greenness. NDVI is expressed as a function of near-infrared radiation and visible radiation: $$\mathrm{NDVI}\;=\;\frac{\mathrm{Near}\;\mathrm{Infrared}\;\mathrm{Radiation}-\mathrm{Visible}\;\mathrm{Radiation}}{\mathrm{Near}\;\mathrm{Infrared}\;\mathrm{Radiation}+\mathrm{Visible}\;\mathrm{Radiation}}$$

NDVI is unitless with value ranging from −1 to +1 for a given pixel. Negative values indicate water features (rivers, coastal water and clouds), values close to zero represent no vegetation, and positive values indicate the existence of green vegetation. A higher value is indicative of greener and denser surface vegetation [19].

In the obtained satellite images, cells containing negative NDVI values were excluded in the calculation of the annual average NDVI because those cells represent water features and would lead to underestimation in measuring greenness. Non-negative NDVI values were aggregated into the district-level for each of the 23 satellite images. By averaging the total for each district, exposure to greenness measured as NDVI was examined as quartiles of the annual NDVI of the 74 districts. Processing and aggregation of NDVI were conducted using ArcGIS 10.3.1 (ESRI 2015, Redlands, CA, USA) software. The processed NDVI images of the seven major cities in 16 October 2009 are shown in Appendix A.

Land-use data was also used for measuring the greenness of the residential districts. Data of the forest area (ha) and forest volume (m^3^) of each district in 2009 were obtained from the Korea Forest Service [20]. For this data, the Korea Forest Service defined “forest” as a “cluster of trees at least 0.1 ha except for farmland and grassland”. District area and the number of residents were obtained from the Korean Ministry of Land, Infrastructure and Transport and the 2010 Korean Census [21]. We combined these data to calculate, forest area per district area (%), forest volume per district area (m^3^/ha), forest area per capita (ha/person), and forest volume per capita (m^3^/person).

### 2.4. Covariates

Covariates were selected as potential confounders based on previous associations with mental illness or greenness: age (10-year interval), sex, education level (uneducated, elementary school, middle school, high school, college or higher), occupation (professional or administrative, office worker, sales or service, agricultural or fisheries, elementary worker, armed forces, others including students, housewife, and unemployed), marital status (single, married, divorced or widowed), annual household income (quartile), body mass index (BMI (kg/m^2^); underweight, <18.5; normal, 18.5–22.9; overweight, 23–24.9; obese, ≥25), smoking status (never, former, current), alcohol consumption (non-drinker, those who drink equal to or less than once a month; drinker, those who drink more than once a month; heavy drinker, for men, those who drink more than once a week and ≥7 shots (5 cans of beer) at a time, for women, those who drink more than once a week and ≥5 shots (3 cans of beer) at a time), physical activity more than or equal to five days a week and more than 30 min per day except for walking (yes, no), and deprivation index.

The deprivation index for each district was calculated using 2010 Korean Census data from the Korean Statistical Information Service in order to adjust for SES at the area level [22]. The deprivation index is defined as the summed z-scored value of proportions [single household, household living in the poor residential environment, female householder, low educated, of the elderly (age ≥ 65), divorced or widowed, households who do not possess a private vehicle, and non-apartment residence]. A higher deprivation index value indicates greater district-level deprivation and lower value indicates lower deprivation.

### 2.5. Statistical Analysis

Logistic regression models were constructed to analyze the association between greenness exposure and depressive symptoms, adjusting for potential confounders. Odds ratios (OR), with 95% confidence intervals (CI) for reporting depressive symptoms were estimated with the quartile of the lowest NDVI (quartile 1) as the reference group for the analysis. To evaluate potential confounding, covariates were additionally included in the model a priori based on previous studies [23,24,25]. In the basic model, we adjusted for age and sex. To further evaluate potential confounding by SES, we additionally adjusted for education, occupation, marital status, annual household income, and deprivation index. Further covariates including BMI, smoking status, and alcohol consumption were additionally included to control for physiological factors. Finally, physical activity was included in the last model. Physical activity may be a potential confounder or a mediating factor linking greenness exposure and depressive symptoms. These four models were conducted for NDVI and only fully adjusted model was conducted for the land-use derived greenness measures.

All statistical analyses were conducted using SAS, version 9.4 (SAS Institute Inc., Cary, NC, USA). PROC SURVEYLOGISTIC procedure was used for the analysis. To take into account the stratified two-stage cluster sample design of the CHS, sampling weight which included sampling probability, response rate, and post-stratification was considered for each study subjects to secure the representative estimates for the population.

## 3. Results

Table 1 depicts the demographic characteristics of the study population. The average age was 46.2 years and 53.5% of the study population was female. 71.1% of the study population finished at least high school education. Approximately half of the individuals’ BMIs were normal, ranging from 18.5–23. 62.9% had never smoked before and 45.1% drank less than once a month. About four-fifths of the participants did not partake in any physical activity. The average CES-D score was 6.35 and 11.7% of the study population was shown to have depressive symptoms based on the CES-D score cut-off point of ≥16. The spatial distribution of the deprivation index for the study area is presented in Appendix A.

The characteristics of the study area are summarized in Table 2. The average area of the 74 districts was 7273 (ha). According to the 2010 Korea Census, the average number of population was 302,410. The annual average NDVI for the 74 districts ranged from 0.20 to 0.61 and with a mean value of 0.38. The average proportion of forest area was 34.26%. The minimum values of forest area and forest volume for the 74 districts were zero as one district did not contain a forest large enough to be defined as a forest by the Korea Forest Service (0.1 ha). The spatial distribution of annual average NDVI for the study area is shown in Figure 1.

The correlations between greenness indicators for all districts combined are shown in Appendix A. NDVI had higher positive correlations with forest area per district area (%), and forest volume per district area (m^3^/ha) with Pearson correlation coefficients of 0.94 (*p*-value < 0.001) and 0.94 (*p*-value < 0.001), respectively. Among the greenness indicators, the correlation of forest area (ha) and forest volume (m^3^) was the highest with Pearson correlation coefficients of 0.99 (*p*-value < 0.001), followed by that of forest area per district area (%) and forest volume per district area (m^3^/ha) with r = 0.93 (*p*-value < 0.001), and forest area per capita (ha/person) and forest volume per person (m^3^/person) with r = 0.93 (*p*-value < 0.001).

Table 3 shows the odds ratios (ORs) for reporting depressive symptoms (CES-D score ≥ 16) in the quartiles for greenness by NDVI, with individuals in the lowest greenness quartile (quartile 1) as the reference group. We used incremental models to assess how the association between greenness exposure and depressive symptoms changed with the inclusion of SES and health behavior variables. Analyses showed a consistent association between higher greenness and lower rate of depressive symptoms, robust to adjustment for potential confounders. In Model 1, when adjusted for age and sex, those living in the greener area (quartiles 2, 3, and 4) had lower odds of having depressive symptoms with ORs of 0.857 (95% CI: 0.788, 0.932), 0.972 (95% CI: 0.893, 1.058), and 0.813 (95% CI: 0.747, 0.884), respectively. When education, job, marital status, household income, and deprivation index were additionally included in Model 2 to adjust for individual level and area level of SES, a reduction in OR was found for all quartiles. Those living in the highest quartile of greenness had a 23.6% lower rate of odds of having depressive symptoms than those in the lowest quartile (OR = 0.764; 95% CI: 0.701, 0.832). In Model 3, additionally adjusted for BMI, smoking status, alcohol consumption, the results were almost unchanged with the OR in quartile 4, at 0.764 (95% CI: 0.702, 0.833). Finally, physical activity was additionally adjusted for in Model 4 to evaluate whether physical activity could be a confounder or a mediating factor. The results remained unchanged with the OR in quartile 4, at 0.765 (95% CI: 0.702, 0.833). In all of those models, individuals in quartile 4 had the lowest OR, compared to those in other quartiles.

Figure 2 shows the fully adjusted ORs of reporting depressive symptoms in the quartiles for urban greenness by each greenness indicator. For all greenness indicators except for forest area per district area (%), the highest rate of depressive symptoms was found for the individuals in the lowest quartile of greenness (quartile 1) and the lowest rate of depressive symptoms for those in the highest quartile of greenness (quartile 4). Linear trends across the greenness quartiles were found for forest volume (m^3^), forest area per capita (ha/person), and forest volume per capita (m^3^/person) with the lowest ORs in quartile 4, at 0.826 (95% CI: 0.757, 0.901), 0.874 (95% CI: 0.800, 0.954), and 0.794 (95% CI: 0.729, 0.865), respectively. The trends of the ORs for NDVI, forest area (ha), and forest volume per district area (m^3^/ha) were similar with the higher ORs in quartile 3 at 0.927 (95% CI: 0.851, 1.010), 0.942 (95% CI: 0.862, 1.030), and 0.932 (95% CI: 0.856, 1.015), respectively. All of the greenness indicators showed the lowest ORs in quartile 4, except for forest area per district area (%) with the lowest OR in quartile 3, at 0.879 (95% CI: 0.808, 0.956). Of those ORs in quartile 4, NDVI showed the lowest OR of 0.765 (95% CI: 0.702, 0.833), followed by 0.794 (95% CI: 0.729, 0.865) in forest volume per capita (m^3^/person).

## 4. Discussion

We investigated the association between urban greenness and depressive symptoms of adults in the seven major cities of Korea. Higher level of greenness exposure was associated with lower rate of depressive symptoms regardless of adjustment for age, sex, education level, occupation, marital status, household income, BMI, smoking status, alcohol consumption, physical activity, and deprivation index. In our analysis, urban greenness was quantified using both NDVI and land-use data. A negative association between greenness exposure and depressive symptoms was found for those various greenness indicators. All of the greenness indicators in our study showed the highest rate of depressive symptoms in the lowest quartile of greenness (quartile 1) and the lowest rate of depressive symptoms for those in the highest quartile of greenness (quartile 4), except for forest area per district area (%).

Our findings were consistent with those of the studies that have examined the association between greenness and depressive symptoms. A negative association between greenness and depressive symptoms was found for Korean general adults when greenness was estimated with green area per capita (ha/person) for each district [25]. Reklaitiene et al. measured depressive symptoms through CES-D 10 which is a short version of CES-D 20, and found an inverse association between the use of the park, residential proximity to the park, and depressive symptoms among 45–72 years old residents in Kaunas (Lithuania) [26]. Our findings differed from those of the several studies. A study in Korea examined the association between community environmental factors, and stress and depressive symptoms in Seoul but found no consistent evidence for greenness exposure [27]. Those environmental factors included greenness area, different types of park area per capita, and the number of green facilities. Another study which used CES-D 20 with a cut-off score of ≥16 suggested an independent relationship between the number of trees, house with private gardens, and depressive symptoms among adults in London (UK) [28]. Differences in study designs and measurements of greenness could explain inconsistent results.

Compared to other studies on the prevalence of depressive symptoms among Korean adults using CES-D, the point prevalence of depressive symptoms of our study population (11.7%) was relatively lower. According to Cho et al., 25.3% of the study population in 1994 was found to have depressive symptoms by the same criteria. In the 2005 study by Kim et al., 38.3% of the adults in 1999 and 2000 were found to have depressive symptoms by the same criteria. Several factors including drastic social and cultural changes due to rapid industrialization which began from the 1960s, and Korean financial crisis of 1997 were expected to result in a higher prevalence of depressive symptoms among the Korean population in the 1990s [29,30]. The prevalence of depressive symptoms in Korean population by CES-D seemed to have decreased compared to the 1990s, however, depression still remains a severe problem in Korea.

Epidemiological studies have assessed residential greenness with various measurements. With advances in the accessibility of satellite-derived data, epidemiological studies assessing greenness by satellite imagery have been actively published since 2010. James et al. examined the association between residential greenness and mortality using NDVI obtained from MODIS [31]. Son et al. also used NDVI from MODIS to assess the effect modification of greenness on heat-related mortality [32]. Several studies measured the association between urban greenness and mental health using NDVI obtained from Landsat satellites [13,33,34]. NDVI is a consistent and comparable greenness measurement for the whole world because it is a satellite-derived data provided for the entire earth. Moreover, NDVI captures small-scale of urban vegetation such as grass and street trees that land-use data cannot. However, NDVI data cannot be used to determine the type of green space and accessibility to the green space. Land-use data typically contains specific information about the type of green space such as parklands and farmlands but lacks the same temporal variation and global standardization that a satellite sensor provides.

In this study, greenness was assessed by NDVI as well as a variety of greenness indicators derived from land-use data. The correlations between NDVI and the greenness indicators derived from the land-use data were positive with NDVI having a significantly high Pearson correlations (r = 0.94) with both forest area per district area (%) and forest volume per district area (m^3^/ha). This may be because NDVI indicates the amount of vegetation in a given cell. Although the results of the constructed models were slightly different according to the greenness indicators, the inverse relationship between urban greenness and depressive symptoms was found to be consistent for those indicators. Since results may vary depending on which greenness measurement is used, using divergent greenness assessments should be accounted for when examining the health impact of greenness.

In our study, we additionally adjusted for physical activity in the final model because this variable may be a confounder or a mediator for the association between greenness exposure and depressive symptoms. The fact that greenness exposure remained associated with depressive symptoms after additionally adjusting for physical activity suggests that alternative pathways may link greenness exposure to depressive symptoms. The direct impact of greenness on mental health may provide a potential explanation of this result. The restorative effect of greenness by itself could be able to offer health benefits without indirect pathways [8,35]. Moreover, social cohesion, air pollution, and noise could be one of the possible indirect pathways that we did not take into account in our analysis. Sugiyama et al. found that perceived greenness remained as an independent predictor of mental health after adjusting for walking and suggested restorative effects of greenness as a potential factor explaining the pathway for health benefits of greenness [36]. Hystad et al. observed no attenuation of the association between greenness exposure and preterm birth when adjusted for air pollution, noise, and distance to the green space, and suggested alternative pathways including psychological and psychosocial influence [37]. Several studies conducted mediation analysis for greenness exposure and health. James et al. found that the association between greenness and mortality could be partially mediated by physical activity, air pollution, social cohesion, and depression [31]. De Vries et al. suggested that the positive effect of greenness exposure on health is mediated by stress and social cohesion, rather than its direct impact on health [38]. Zijlema et al. expected social cohesion, physical activity, loneliness, mental health, air pollution worries, and noise as mediators for the relationship between natural outdoor environment and cognitive functioning but none of them were found to be mediating factors [39]. More studies are needed to explore the possible mechanisms of the health benefit of greenness exposure.

One of the strengths of our study is that we used national survey data (CHS 2009) which offers representativeness of the general population in South Korea. We compared our study population with that of the Korea National Health and Nutrition Examination Survey 2009, and the characteristics of those two populations including age, sex, education level, occupational classification and marital status were similar. Moreover, this is the first study which utilized NDVI data as well as land-use data to examine the mental health benefits of urban greenness in Korea. We were able to take into account various greenness indicators in our analysis for comparison.

As for limitations, the absence of residential history might have introduced bias due to exposure misclassification. There might be some individuals who recently moved in from other areas. However, we assumed that the proportion of the incoming individuals would not be considerably different depending on the areas. If there has been differential exposure misclassification, it would be the case that the individuals’ choice of their residence is related to their mental health status. However, in Korea, especially for the major cities, financial condition and accessibility to public transportation are the major factors for choosing a residential area. Therefore, the individuals’ mental health status might not have had a big influence on the choice of residence and we could expect that the degree of misclassification was non-differential and our results might have been underestimated. Second, compared to the individuals who got depressed recently, some might have had long-term depression and these people could have brought bias to our analysis. However, the proportion of these people would be randomly distributed in the districts regardless of greenness level and non-differential misclassification would have occurred. Third, because the study participants’ residential locations were only available at the district-level, we were not able to assess exposure of each individual’s residential greenness at a higher resolution. However, greenness at the district-level could still be a valid measurement for assessing an individual’s exposure to greenness. People could be exposed to greenness outside of their residence through commuting or doing physical activities in a nearby park. Moreover, our study lacks the information on accessibility to green space. But we assumed that the greenness level we measured reflects the general accessibility to urban greenness. Finally, this study did not take into account the impact of air pollution and noise which could lead to residual confounding. Further research on health benefit of greenness exposure that includes these environmental risk factors of mental health is needed. Despite the limitations, to our knowledge, this is the first study which aimed at measuring the effect of urban greenness on mental health with various greenness indicators including NDVI.

Among the various causes of depression, environmental factors have received less attention compared to other risk factors. As depressive disorders result from a complex interaction of many factors, a multi-dimensional approach to depression is needed [40]. Our study suggests that urban greenness may be beneficial for depressive symptoms with consistent results from a variety of greenness indicators. Further study, especially prospective analyses which include information on depression onset and consideration on the stages of life course where the impact of greenness exposure could be critical will help to understand the causal relationship of greenness exposure and mental health [41,42].

## 5. Conclusions

In conclusion, our study suggests the protective impact of urban greenness on the depressive symptoms. The results were consistent for various greenness indicators derived from satellite imagery and land-use data. Our analyses could provide a scientific basis for urban planning and policy making for public health promotion. Further studies are needed for understanding the impact of environmental factors on mental health.

## Figures and Tables

**Figure 1 ijerph-16-00173-f001:**
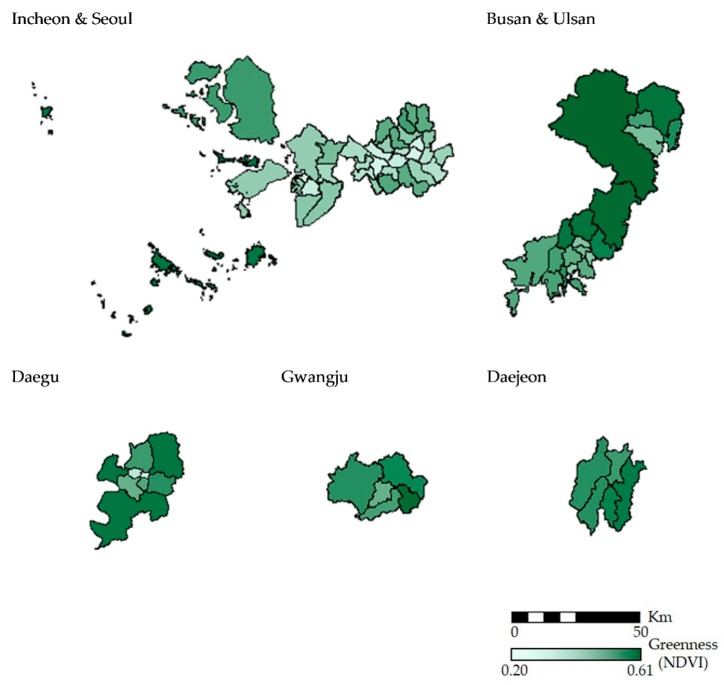
The spatial distribution of greenness level (annual average NDVI) of the study area.

**Figure 2 ijerph-16-00173-f002:**
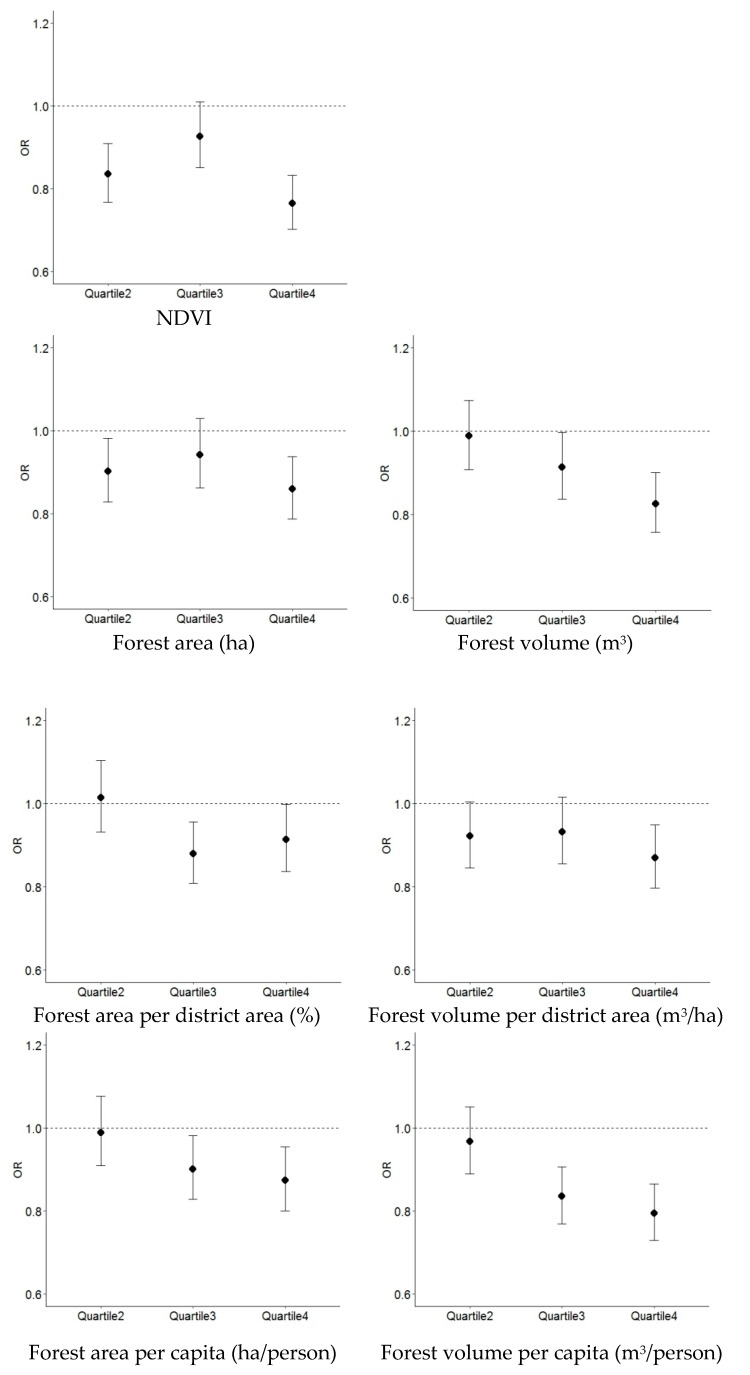
Fully adjusted ORs for reporting depressive symptoms in the quartiles for urban greenness by various greenness indicators. NDVI: Normalized Difference Vegetation Index.

**Table 1 ijerph-16-00173-t001:** Characteristics of the study population.

Variables		N	%	Variables		N	%
Participants		65,128	100.0	BMI	Underweight	3993	6.1
Age (years)	Mean (SD)	46.2 (15.9)			Normal	31,457	48.3
Sex	Male	30,300	46.5		Overweight	15,483	23.8
	Female	34,828	53.5		Obese	14,195	21.8
Education level	Uneducated	3949	6.1	Smoking status	Never	40,977	62.9
	Elementary	7318	11.2		Former	8523	13.1
	Middle school	7541	11.6		Current	15,628	24.0
	High school	25,089	38.5	Alcohol consumption	Non-drinker	29,377	45.1
	College or higher	21,231	32.6				
Annual household-adjusted income	<25th (0–9809 $)	16,210	24.8		Drinker	28,179	43.3
	25th–50th (9862–15,402 $)	16,719	25.7		Heavy drinker	7572	11.6
	50th–75th (15,474–22,232 $)	16,063	24.7	Physical activity	No	52,195	80.1
	>75th (22,410–53,357 $)	16,136	24.8		Yes	12,933	19.9
Marital status	Single	13,876	21.3	CES-D score	<16	57,512	88.3
	Married	41,395	63.6		≥16	7616	11.7
	Divorced or widowed or separated	9857	15.1		Mean (SD)	6.35 (7.9)	

**Table 2 ijerph-16-00173-t002:** Characteristics of the districts (*N* = 74).

Characteristic	Mean (SD)	Min	25th	Median	75th	Max
District area (ha)	7273 (11,303)	282	1,760	35.69	6845	75,534
Number of population ^a^	302,410 (152,205)	14,550	169,814	303,686	410,084	646,970
NDVI	0.38 (0.11)	0.20	0.29	0.39	0.47	0.61
Forest area (ha)	3511 (7329)	0	329	1109	3474	52,306
Forest volume (m^3^)	358,319 (715,470)	0	32,157	106,911	326,448	4,988,441
Forest area per district area (%)	34.26 (19.89)	0	18.03	33.07	50.03	71.11
Forest volume per district area (m^3^/ha)	35.53 (23.44)	0	15.98	36.05	50.45	102.89
Forest area per capita (ha/person)	3.10 (10.54)	0	0.14	0.33	1.04	79.76
Forest volume per capita (m^3^/person)	2.84 (8.51)	0	0.11	0.36	1.10	61.63

^a^ 2010 Korean Census data.

**Table 3 ijerph-16-00173-t003:** Odds ratios for reporting depressive symptoms in the quartiles for urban greenness by NDVI.

NDVI	Odds Ratio (95% Confidence Interval)			
	Model 1	Model 2	Model 3	Model 4
Quartile 1 (0.20–0.29)	Reference	Reference	Reference	Reference
Quartile 2 (0.30–0.39)	0.857 (0.788, 0.932)	0.829 (0.762, 0.903)	0.836 (0.768, 0.910)	0.836 (0.768, 0.910)
Quartile 3 (0.40–0.45)	0.972 (0.893, 1.058)	0.927 (0.851, 1.010)	0.927 (0.851, 1.010)	0.927 (0.851, 1.010)
Quartile 4 (0.47–0.61)	0.813 (0.747, 0.884)	0.764 (0.701, 0.832)	0.764 (0.702, 0.833)	0.765 (0.702, 0.833)

Model 1: Adjusted for age (10-years category), sex, Model 2: Model 1 + education, job, marital status, household income, deprivation index, Model 3: Model 2 + BMI, smoking status, alcohol consumption, Model 4: Model 3 + physical activity.

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
