# Peer review of "Association between Urban Greenness and Depressive Symptoms: Evaluation of Greenness Using Various Indicators"

_ijerph, 2019, doi:10.3390/ijerph16020173_

Round 1

Reviewer 1 Report

The limitations of the study mentioned in lines 314-318 make this study quite inconclusive. 
In the context of mental health it seems extremely important to know what was the long-term exposure of an individual to a certain environment. As authors noted, we don't know how many of the people in the study were new to the areas/cities. They could have lived in a forest their whole life and moved to a city a year before. This is the "life course" element - in my opinion a major factor missing in the study. This factor is a foundation of similar research, namely:

Cherrie, M.P.C., Shortt, N.K., Mitchell, R.J., Taylor, A.M., Redmond, P., Ward Thompson, C., Starr, J.M., Deary, I.J. and Pearce, J. 2017. ‘
Green space and cognitive ageing: A retrospective life course analysis in the Lothian Birth Cohort 1936′, Social Science & Medicine 196, 56-65
and
https://doi.org/10.1016/j.socscimed.2017.10.038Pearce, J., Cherrie, M., Shortt, N., Deary, I. & Ward Thompson, C. (2018). Life course of place: a longitudinal study of mental health and place. Transactions of the Institute of British Geographers 2018; 00: 1–18. https://doi.org/10.1111/tran.12246

Studies have shown how important an exposure to greenery is at various stages of life. Exposure to nature and "urban greenness" at an early (childhood) stage could be vital to the mental health in the later stages of life. 
Analysing only one moment in persons life in the context of his/hers mental health and the environment that she/he lives in seems to be a huge simplification. 
In my opinion the article presents a efficient statistical analysis of insufficient data, therefore making the research incomplete, inadequate.

Author Response

To the reviewer #1

Dear reviewer

We appreciate your comments on our manuscript entitled “Association between urban greenness and depressive symptoms: evaluation of greenness using various indicators”. All of your comments were very helpful for revising and improving our paper. We have studied these comments carefully and have made corresponding corrections that we hope will meet with your approval. Your comments and our responses are provided below.

We would like to express our great appreciation to you and you for the comments on our paper. If you have any further queries, please do not hesitate to contact us.

Kind regards,

Jong-Tae Lee

The limitations of the study mentioned in lines 314-318 make this study quite inconclusive. In the context of mental health, it seems extremely important to know what was the long-term exposure of an individual to a certain environment. As authors noted, we don't know how many of the people in the study were new to the areas/cities. They could have lived in a forest their whole life and moved to a city a year before. This is the "life course" element - in my opinion a major factor missing in the study. This factor is a foundation of similar research, namely:

Cherrie, M.P.C., Shortt, N.K., Mitchell, R.J., Taylor, A.M., Redmond, P., Ward Thompson, C., Starr, J.M., Deary, I.J. and Pearce, J. 2017. ‘Green space and cognitive ageing: A retrospective life course analysis in the Lothian Birth Cohort 1936′, Social Science & Medicine 196, 56-65

and

https://doi.org/10.1016/j.socscimed.2017.10.038Pearce, J., Cherrie, M., Shortt, N., Deary, I. & Ward Thompson, C. (2018). Life course of place: a longitudinal study of mental health and place. Transactions of the Institute of British Geographers 2018; 00: 1–18. https://doi.org/10.1111/tran.12246

Studies have shown how important an exposure to greenery is at various stages of life. Exposure to nature and "urban greenness" at an early (childhood) stage could be vital to the mental health in the later stages of life.

Analysing only one moment in persons’ life in the context of his/hers mental health and the environment that she/he lives in seems to be a huge simplification.

In my opinion the article presents an efficient statistical analysis of insufficient data, therefore making the research incomplete, inadequate.

In our analysis, we used one-year average greenness and we assumed that it could be representative for the exposure of much longer period. We recently finished gathering NDVI data from 2006 to 2015 and found that greenness level of a district did not change drastically over time (The Pearson correlation coefficients of the annual average NDVI values from 2006 to 2015 were higher than 0.98 for all 74 districts). Therefore, we expected that we could estimate the longer period of greenness exposure through one-year average NDVI.

However, there could be some individuals who have lived in a greener area for a long time and recently moved to less green area. Due to these incoming people, misclassification could have occurred for our use of longer-term exposure of greenness. However, the proportion of the incoming individuals would not be considerably different across the different districts. If differential exposure misclassification occurred in our analysis, it could be the case that the individuals’ choice of their residence is related to their mental health status. However, in Korea, especially for the major cities, financial condition and accessibility to public transportation are the major factors for choosing a residential area. Therefore, the individuals’ mental health status might not have influenced the choice of residence. We could expect that the degree of misclassification was non-differential and our results might have been underestimated. In conclusion, even with those incoming individuals, we could assess longer-term greenness exposure through one-year average NDVI and validly assess the association between greenness exposure and depressive symptoms.

Moreover, we agree that analyzing only a short period in the individuals’ life could be a simplification in the context of mental health and the environment. Compared to individuals who recently got depressed, some might have had depression for a long time and these people could have brought bias into our analysis. However, the proportion of these people would be randomly distributed in the districts regardless of greenness level and non-differential misclassification would have occurred.

We also agree that even though we could estimate long-term greenness through one-year average greenness, we were not able to consider the effect of exposure to nature on one’s health at various life stages.

 Therefore, our study has those limitations due to the cross-sectional data. However, many cross-sectional studies have published meaningful results.

Beyer KM, Kaltenbach A, Szabo A, Bogar S, Nieto FJ, Malecki KM. Exposure to neighborhood green space and mental health: evidence from the survey of the health of Wisconsin. Int J Environ Res Public Health. 2014;11(3):3453–72.2.

Triguero-Mas M, Dadvand P, Cirach M, Martinez D, Medina A, Mompart A, et al. Natural outdoor environments and mental and physical health: relationships and mechanisms. Environ Int. 2015;77C:35–41.

Astell-Burt T, Feng X, Kolt GS. Mental health benefits of neighbourhood green space are stronger among physically active adults in middle-to-older age: evidence from 260,061 Australians. Prev Med. 2013;57(5):601–6.

Fan Y, Das KV, Chen Q. Neighborhood green, social support, physical activity, and stress: assessing the cumulative impact. Health Place. 2011;17(6):1202–11.

Especially, because studies on the association between greenness exposure and mental health are very limited in Korea, we would contend that our study is a meaningful attempt even with those limitations. We are currently on the process of analyzing the health impact of greenness exposure with longitudinal data and hope to publish more valid and convincing studies.

 Thanks to your valuable comments, we could make revisions to the Discussion (lines 334-346 and line 362-365). We hope the edited section clarifies the limitations properly.

“For limitations, the absence of residential history might have introduced bias due to exposure misclassification. There might be some individuals who recently moved in from other areas. However, we assumed that the proportion of the incoming individuals would not be considerably different depending on the areas. If there has been differential exposure misclassification, it would be the case that the individuals’ choice of their residence is related to their mental health status. However, in Korea, especially for the major cities, financial condition and accessibility to public transportation are the major factors for choosing a residential area. Therefore, the individuals’ mental health status might not have had a big influence on the choice of residence and we could expect that the degree of misclassification was non-differential and our results might have been underestimated. Second, compared to the individuals who got depressed recently, some might have had long-term depression and these people could have brought bias to our analysis. However, the proportion of these people would be randomly distributed in the districts regardless of greenness level and non-differential misclassification would have occurred.”

 “Further study, especially prospective analyses which include information on depression onset and consideration of the stages of life course where the impact of greenness exposure could be critical will help to understand the causal relationship of greenness exposure and mental health [Cherrie et al., Pearce et al.].”

Reviewer 2 Report

Originality/Novelty: Is the question original and well defined? Do the results provide an advance in current knowledge?

Article contributes additional knowledge, once issues are resolved.  Article needs copy edit for English word usage.  For example use age of the word “decrease” in Odds Ratio implies that findings are longitudinal, which they are not.

Significance: Are the results interpreted appropriately? Are they significant? Are all conclusions justified and supported by the results?  Are hypotheses and speculations carefully identified as such?

Authors should more clearly and state the specific hypotheses tested and whether they are accepted or rejected. Readers need and would appreciate a more detailed explanation of the analytic models developed and tested.  (Pages 7-8).

Quality of Presentation: Is the article written in an appropriate way? Are the data and analyses presented appropriately? Are the highest standards for presentation of the results used?

Regarding the use of MODIS for the NDVI/Greenness measure. Please explain why MODIS with a rather coarse 250m resolution was selected for use in these urban areas, as opposed to other available imagery that his much higher resolution.

Please include a Table describing the areas of the MODIS Categories as well as the Forest amount and mass for each urban area and for study as a whole.  This would also help in understating the results when the NDVI quartiles are presented and are quite important in the discussion.

Readers also need to understand some basic characteristics of the urban areas studied.  The maps in Figure 1 are a beginning, but please includes some basic statistics such as population in the cities and in the districts and the area of each.

An analytic question is why the individual urban areas were also not analyzed separately. It is clear in Figure 1 that some cities have much more green than others.Is this important?  Also, these areas appear to be fairly large (more that 25km in at least one dimension) and it may be the case that the district boundaries themselves affect the amount of greenness and hence the results. Please discuss.  Perhaps a multi-level model could be employed.

Regarding correlations between NDVI and Forest, (Table 2), what explains the differences between the two data sources? Are those differences important?

Table 1 should definitely include descriptive statistics for the Deprivation Index.

Please include all topics employed in the Discussion in the Introduction.

Scientific Soundness: is the study correctly designed and technically sound? Are the analyses performed with the highest technical standards? Are the data robust enough to draw the conclusions? Are the methods, tools, software, and reagents described with sufficient details to allow another researcher to reproduce the results?

Needs more details on the formal models and hypotheses and why this particular statistical approach was employed, as well as the above-mentioned justification for using MODIS.

I would appreciate a deeper discussion of the “Protective Impact” finding.

A significant limitation is that there no information of the location of respondents with respect to the green areas. This seems important to discus, considering the large size of these urban areas  

Author Response

To the reviewer #2

Dear reviewer

We appreciate your comments on our manuscript entitled “Association between urban greenness and depressive symptoms: evaluation of greenness using various indicators”. All of your comments were very helpful for revising and improving our paper. We have studied these comments carefully and have made corresponding corrections that we hope will meet with your approval. Your comments and our responses are provided below.

We would like to express our great appreciation to you and you for the comments on our paper. If you have any further queries, please do not hesitate to contact us.

Kind regards,

Jong-Tae Lee

Originality/Novelty: Is the question original and well defined? Do the results provide an advance in current knowledge?

Article contributes additional knowledge, once issues are resolved. Article needs copy edit for English word usage. For example, use age of the word “decrease” in Odds Ratio implies that findings are longitudinal, which they are not.

Thank you for your comment. We agree that findings are not longitudinal and usage of the word “decrease” is not correct in this case. We changed “decreased rate of odds” into “lower rate of odds”, and “largest decrease in odds” into “lowest OR”. (line 224, 230, 237, 241-243)

Significance: Are the results interpreted appropriately? Are they significant? Are all conclusions justified and supported by the results?  Are hypotheses and speculations carefully identified as such?

Authors should more clearly and state the specific hypotheses tested and whether they are accepted or rejected. Readers need and would appreciate a more detailed explanation of the analytic models developed and tested. (Pages 7-8).

 We added a clearer statement of our hypotheses in the Introduction and the Result (line 70-73, line 213-230). We also added more detailed explanation on the statistical models in the Result (line 213-230).

Quality of Presentation: Is the article written in an appropriate way? Are the data and analyses presented appropriately? Are the highest standards for presentation of the results used?

Regarding the use of MODIS for the NDVI/Greenness measure. Please explain why MODIS with a rather coarse 250m resolution was selected for use in these urban areas, as opposed to other available imagery that his much higher resolution.

We used MODIS on board NASA's Terra satellite which is a commonly used measure of exposure to neighborhood greenness that has been used widely to examine health impacts. We agree that there are more highly resolved satellites such, however since our exposure is assigned at the district level using a higher resolution image such as from LandSat which is available to 30 m would still need to be averaged a larger spatial scale as were the MODIS values which reduces the fine-scale spatial variance when using a higher resolution image.

Please include a Table describing the areas of the MODIS Categories as well as the Forest amount and mass for each urban area and for study as a whole. This would also help in understating the results when the NDVI quartiles are presented and are quite important in the discussion.

 Thank you for the suggestion. We have added a new table (Table 2) presenting the information on NDVI, forest area, and forest volume.

Readers also need to understand some basic characteristics of the urban areas studied.  The maps in Figure 1 are a beginning, but please include some basic statistics such as population in the cities and in the districts and the area of each.

We included basic statistics of the study area such as district area and number of population based on 2010 Korean Census data (Table 2).

An analytic question is why the individual urban areas were also not analyzed separately.  It is clear in Figure 1 that some cities have much more green than others. Is this important? Also, these areas appear to be fairly large (more that 25km in at least one dimension) and it may be the case that the district boundaries themselves affect the amount of greenness and hence the results. Please discuss. Perhaps a multi-level model could be employed.

We perceive your comment to mean that if the boundary of a district is large, the amount of greenness would be large as well. However, NDVI indicates the amount of vegetation in a given pixel (250m2) and NDVI value of a district means the average value of NDVIs in the pixels of the district. Therefore, NDVI indicates the quality of greenness in the district and the case that the district boundaries themselves affect the amount of greenness would be not likely. This also could be the same for the greenness indicators such as forest area per district area and forest volume per district area. Those variables indicate the proportion of green space (%) and amount of greenness per area (m3/ha).

Regarding correlations between NDVI and Forest, (Table 2), what explains the differences between the two data sources? Are those differences important?

 We apologize for the confusion. The differences between those two data sources were explained in the methods and the discussion (line 98-127 and line276-288). As we have added a new table describing the characteristics of the greenness indicators, we moved the table presenting correlations between the greenness indicators to the Supplemental materials (Table S1).

Table 1 should definitely include descriptive statistics for the Deprivation Index.

 Thank you for your comment. However, because the Deprivation Index is z-scored values for the districts, presenting the Deprivation Index in the table might not be informative. Instead, we added a map presenting the index (Figure S2).

Please include all topics employed in the Discussion in the Introduction.

 We apologize for the less organized Introduction. We reorganized the Introduction and the Discussions in a clearer way and added relevant references.

Scientific Soundness: is the study correctly designed and technically sound? Are the analyses performed with the highest technical standards? Are the data robust enough to draw the conclusions? Are the methods, tools, software, and reagents described with sufficient details to allow another researcher to reproduce the results?

Needs more details on the formal models and hypotheses and why this particular statistical approach was employed, as well as the above-mentioned justification for using MODIS.

 As we mentioned above, we added more detailed explanation on the statistical models and hypotheses in the Introduction and the Results.

I would appreciate a deeper discussion of the “Protective Impact” finding.

 We appreciate your comment. We added detailed discussions on the main findings. (line 248-257)

A significant limitation is that there no information of the location of respondents with respect to the green areas. This seems important to discus, considering the large size of these urban areas 

 We agree that the absence of the information on the exact location of respondents is one of the significant limitations. Many studies have measured greenness in a higher resolution such as 300m or 1km radial buffer of residence. In our study, exposure of each individual’s residential greenness could have been biased due to the large size of the districts. However, if we consider the region where individuals travel for their daily lives, district-level greenness also could be a valid measurement for assessing the association between greenness exposure and its health impact. For example, people could be exposed to greenness during commuting, or people could go to parks near their residences for physical activity. Greenness exposure during these activities also could be beneficial for their mental health. Therefore, even with relatively coarse resolution of greenness exposure due to the limitation of the data, district-level greenness also could be a valid measurement for assessing the association between greenness exposure and its health impact.

 We appreciate your valuable comment on the limitation and we have included sentences about it. (line 346-351)

 “Third, because the study participants’ residential locations were only available at the district-level, we were not able to assess exposure of each individual’s residential greenness at a higher resolution. However, greenness at the district-level could still be a valid measurement for assessing an individual’s exposure to greenness. People could be exposed to greenness outside of their residence through commuting or doing physical activities in a nearby park.”

Reviewer 3 Report

This was a straightforward but interesting paper offering a new insight into greenness and mental health in Korea. Everything that has been done is sound to me, it reads quite well. I have some small comments below listed by line, I have also added a suggestion for further analysis at the end of my review. It would extend the paper quite nicely I think but it is not necessary to complete.

Minor comments

Line 42 - missing word?

Lines 51-53 are quite repetitive of lines 43-45 - can you edit?

Line 56 - you reference several papers that look at different mental health issues - but it might be better to narrow down to depression. For example, see McEachan, R.R.C., Prady, S.L., Smith, G., Fairley, L., Cabieses, B., Gidlow, C., Wright, J., Dadvand, P., Van Gent, D. and Nieuwenhuijsen, M.J., 2016. The association between green space and depressive symptoms in pregnant women: moderating roles of socioeconomic status and physical activity. J Epidemiol Community Health70(3), pp.253-259, and also see Helbich, M., Klein, N., Roberts, H., Hagedoorn, P., & Groenewegen, P. P. (2018). More green space is related to less antidepressant prescription rates in the Netherlands: A Bayesian geoadditive quantile regression approach. Environmental Research166, 290-297.

Line 92 - is there a reference for this?

Line 141 - "those who drink more than drinkers do" - can you be more specific? this is quite a confusing definition

Table 1 - I don't really follow what the numbers in the income range? This is the currency? It looks a bit odd under the N and % heading at the top. Maybe you could relabel the <35, 25-50 etc names?

Since most of the information on how the data was categorised is given in the methods section, I don't see the point in repeating it all in the table note.

Line 186 - the table says 53.5%? Which one is right

Line 189 - it might be worth a small comment on what the 6.35 CES-D score means in terms of the cut off point. Maybe also small comment on the study population % - is this what you expected? Is it in line with previous research in Korea?

Line 200 - do you mean the per capita results? You could talk about the results a little more - at the moment quite succinct!

Table 3 - heading in bold? Also, I think you should indicate which quartile had the most green and least green to the table, since I forgot and had to go back and check. It would help the reader interpret the table faster. (This is also applicable to Figure 2).

Line 237-238 - please give the main results in more detail.

Line 257-268 - You discuss here the use of NDVI, but mainly talk about its advantages over land-use data. can you give some disadvantages? no indication of access to green space or quality of it?

Line 272 - shouldn't it be (r=0.94) ?

Line 277 - "for when examining"

Line 278-284 - this seems to be quite out of place. Could you integrate it with the paragraph at lines 247-256?

Line 290-291 - insert brief reference to Ulrich and Kaplan+Kaplan theories?

Line 292 - one of the possible pathways - doesn't make sense

Line 322 - change 'supposed' to 'assumed'

Statistical comment

You mention a few times in the manuscript that physical activity can be a confounder or a mediator for the relationship between greenness and depression. I found it quite surprising that you did not complete a mediation analysis to test this relationship, given you have the data. I think it is possible to still do when you have a dichotomous outcome variable? 

Did you also check any potential interaction effects? By education or occupation maybe? There are several studies that have indicated that the relationship between green space and health is stronger for those who are with less education or lower social class. 

I think it would make a stronger paper if you check for moderators and mediators, but I will not make it a necessary thing to complete. Just a thought!

Author Response

To the reviewer #3

Dear reviewer

We appreciate your comments on our manuscript entitled “Association between urban greenness and depressive symptoms: evaluation of greenness using various indicators”. All of your comments were very helpful for revising and improving our paper. We have studied these comments carefully and have made corresponding corrections that we hope will meet with your approval. Your comments and our responses are provided below.

We would like to express our great appreciation to you and you for the comments on our paper. If you have any further queries, please do not hesitate to contact us.

Kind regards,

Jong-Tae Lee

Minor comments

Line 42 - missing word?

Thank you for your comment. We have revised “one of the important environmental” into “one of the most important environmental”. (line 39)

Lines 51-53 are quite repetitive of lines 43-45 - can you edit?

 We appreciate your comment. We revised that part more concisely (line39-47).

Line 56 - you reference several papers that look at different mental health issues - but it might be better to narrow down to depression. For example, see McEachan, R.R.C., Prady, S.L., Smith, G., Fairley, L., Cabieses, B., Gidlow, C., Wright, J., Dadvand, P., Van Gent, D. and Nieuwenhuijsen, M.J., 2016. The association between green space and depressive symptoms in pregnant women: moderating roles of socioeconomic status and physical activity. J Epidemiol Community Health, 70(3), pp.253-259, and also see Helbich, M., Klein, N., Roberts, H., Hagedoorn, P., & Groenewegen, P. P. (2018). More green space is related to less antidepressant prescription rates in the Netherlands: A Bayesian geoadditive quantile regression approach. Environmental Research, 166, 290-297.

We have revised it to narrow down to depression and thank you for suggesting the relevant references (line 47-60).

Line 92 - is there a reference for this?

Thank you for your suggestion. Unfortunately, because the reference was a report written in Korean and we thought it might be not informative, we added references for CES-D itself in the following sentences (line 97-100).

Line 141 - "those who drink more than drinkers do" - can you be more specific? this is quite a confusing definition

We apologize for the confusion. There was a mistake on the sentence and we revised as follows. (line 143-147)

“drinker, those who drink more than once a month; heavy drinker, for men, those who drink more than once a week and ≥ 7 shots (5 cans of beer) at a time, for women, those who drink more than once a week and ≥ 5 shots (3 cans of beer) at a time”

Table 1 - I don't really follow what the numbers in the income range? This is the currency? It looks a bit odd under the N and % heading at the top. Maybe you could relabel the <35, 25-50 etc names?

 We apologize for the confusion. We revised the label and added the number and % of each category (Table 1).

Since most of the information on how the data was categorised is given in the methods section, I don't see the point in repeating it all in the table note.

 We agree that the table note is repetitive and we have deleted it (Table 1).

Line 186 - the table says 53.5%? Which one is right

We apologize for the confusion. There was a mistake on rounding off the number. 53.5% is right. (line 179)

Line 189 - it might be worth a small comment on what the 6.35 CES-D score means in terms of the cut off point. Maybe also small comment on the study population % - is this what you expected? Is it in line with previous research in Korea?

 Discussion on the CES-D score of our study and that of the previous studies in Korea was mentioned (line 272-281).

Line 200 - do you mean the per capita results? You could talk about the results a little more - at the moment quite succinct!

 We apologize for the confusion due to lack of sufficient explanation. We added more detailed explanations on the correlations of the greenness indicators (line 200-207).

Table 3 - heading in bold? Also, I think you should indicate which quartile had the most green and least green to the table, since I forgot and had to go back and check. It would help the reader interpret the table faster. (This is also applicable to Figure 2).

 We have revised the heading in the table. Thank you for your suggestion on presenting quartile. However, we had presented the range of NDVI under each quartile to indicate which quartile had the most green and least green. Plus, we have added a new table presenting the range of greenness for the study area (Table 2). We hope these revisions provide more clear explanation for the greenness level of the quartiles.

Line 237-238 - please give the main results in more detail.

 We appreciate your comment. We added detailed explanations on the main results. (line 248-257)

Line 257-268 - You discuss here the use of NDVI, but mainly talk about its advantages over land-use data. can you give some disadvantages? no indication of access to green space or quality of it?

We have added an explanation on disadvantages of NDVI that it lacks information on the type of greenness and accessibility of greenness is unable to be considered (line282-294).

Line 272 - shouldn't it be (r=0.94)?

 We have revised as you mentioned. (line 298)

Line 277 - "for when examining"

 We have revised as you mentioned. (line 303)

Line 278-284 - this seems to be quite out of place. Could you integrate it with the paragraph at lines 247-256?

 We have added a paragraph about previous studies on the association between greenness exposure and depressive disorders (line 258-271).

Line 290-291 - insert brief reference to Ulrich and Kaplan+Kaplan theories?

 Thank you for your suggestion. We have added those references. (line310-311)

Line 292 - one of the possible pathways - doesn't make sense

We found logical errors in this paragraph and revised as following (line307-313)

“The fact that greenness exposure remained associated with depressive symptoms after additionally adjusting for physical activity suggests that alternative pathways may link greenness exposure to depressive symptoms. The direct impact of greenness on mental health may provide a potential explanation of this result. The restorative effect of greenness by itself could be able to offer health benefits without indirect pathways. Moreover, social cohesion, air pollution, and noise could be one of the possible indirect pathways that we did not take into account in our analysis.”

Line 322 - change 'supposed' to 'assumed'

 We have revised as you mentioned.

Statistical comment

You mention a few times in the manuscript that physical activity can be a confounder or a mediator for the relationship between greenness and depression. I found it quite surprising that you did not complete a mediation analysis to test this relationship, given you have the data. I think it is possible to still do when you have a dichotomous outcome variable?

Did you also check any potential interaction effects? By education or occupation maybe? There are several studies that have indicated that the relationship between green space and health is stronger for those who are with less education or lower social class.

I think it would make a stronger paper if you check for moderators and mediators, but I will not make it a necessary thing to complete. Just a thought!

 We thank the reviewer for this valuable suggestion on moderation and mediation analysis. We have conducted analysis on the effect modification by air pollution and mediating effect by air pollution, physical activity, and walking. We are currently in the process of writing a paper on it. Again, we appreciate your interest and suggestion for further analysis.

Round 2

Reviewer 1 Report

With the limitations of the study being more carefully explained it should be clearer to the reader exactly what this research has to offer and how it should be further developed. Accepting some of the explanations provided by the authors I would consider publishing the paper in current form.

Reviewer 2 Report

Revised version is acceptable.